# Eating habits of students of health colleges and non-health colleges at the Northern Border University in the Kingdom of Saudi Arabia

**Hanaa E. Bayomy**[1,2]*, **Shmoukh Mushref Alruwaili**[3], **Razan Ibrahim Alsayer**[3], **Nuof Khalid Alanazi**[3], **Dana Ahmed Albalawi**[3], **Khulud Hamed Al Shammari**[3], **Mariam Mahmoud Moussa**[4]

1 Department of Community, Environmental and Occupational Medicine, Faculty of Medicine, Benha University, Benha, Egypt, 2 Department of Family and Community Medicine, Faculty of Medicine, Northern Border University, Arar, Saudi Arabia, 3 Faculty of Medicine, Northern Border University, Arar, Saudi Arabia, 4 Faculty of Medicine, Benha University, Benha, Egypt

* hana.bayuomi@fmed.bu.edu.eg, Hanaa.Sayed@nbu.edu.sa

**Data Availability Statement:** All relevant data are within the paper and its Supporting information files.

## Abstract

### Background

Eating habits are important determinants of health. Young adults who have recently transitioned to university life experience stress and lack of time, which are major risk factors for poor eating habits. This study's objectives were to identify and compare eating habits between students at health and non-health colleges at Northern Border University (NBU), Saudi Arabia and to determine the relationship between students' eating habits and their sociodemographic, lifestyle, and psychological factors.

### Methods

This comparative cross-section study was conducted on 480 students equally from health and non-health colleges at NBU from March to April 2023. A pre-designed electronic questionnaire was used to collect data on students' sociodemographic characteristics, eating habits, and diet-related psychological factors. Statistical analyses were carried out using STATA/SE version 11.2 for Windows.

### Results

Unhealthy eating habits were reported by 44.6% and 41.3% of students at health and non-health colleges, respectively. Most students had irregular meals, skipped breakfast, ate fast food frequently, and consumed insufficient amounts of vegetables, fruits, and water. Parental separation, living away from family predisposed to unhealthy eating habits. Being 21–23 years old, playing sports, and high body mass index were linked to healthy eating habits. Psychological factors such as overeating until the stomach hurts and eating to feel happy were associated with unhealthy eating habits.

**Funding:** The author(s) received no specific funding for this work.

**Competing interests:** The authors have declared that no competing interests exist.

## Conclusion

Unhealthy eating habits were prevalent among students at NBU irrespective of the type of study. Thus, implementing initiatives to promote nutrition and healthy eating habits within the university environment is crucial for health promotion and well-being among students.

## Introduction

Healthy eating habits provide enough micronutrients and hydration to meet the body's physiologic needs. While also supporting the energetic and physiologic needs of the body without overindulging in macronutrients. For example, macronutrients (i.e., carbohydrates, proteins, and fats) provide the energy for cellular processes needed for daily functioning, and micronutrients (i.e., vitamins and minerals), on the other hand, are required in relatively small amounts for normal growth, development, metabolism, and physiologic functioning [1].

Healthy eating also includes consistent food consumption behaviors and patterns, which is very advantageous for supporting and maintaining both physical and psychological health [2]. Furthermore, The World Health Organization (WHO) places a high priority on nutrition and maintaining a healthy diet [3]. Eating habits are the foundation of health as 25–30 percent of our well-being is determined by our eating habits [4].

In addition, the WHO has stated that health-risk behaviors that begin in adolescence (such as unhealthy eating habits) are a major cause of disease burden in adults [5]. For example, most of the Saudi population has higher levels of consumption of refined foods and animal products than fruits and vegetables [6]. Diet-related diseases including cardiovascular disease, and stroke are the leading causes of death worldwide [7]. Moreover, obesity is a consequence of the interaction of poor eating habits, other environmental factors, and genetic predisposition, and plays a fundamental role in the development of coronary heart disease, hypertension, and diabetes mellitus [8].

Many psychological factors affect eating habits and food choices among university students such as stress, depression, boredom, and anxiety [9]. It has been demonstrated that young adults are continually challenged to make healthy food choices after the transition from adolescence to young adulthood when independence increases [10, 11]. However, young adults who have recently made the transition into university life are exposed to stress and a lack of time, which are significant public health causes for poor dietary intake [5]. Medical students have sufficient knowledge about healthy diets and thus are supposed to practice healthier eating habits than non-medical students. However, they rarely apply this knowledge [12]. Additionally, due to the dedication to their studies and clinical rotations in their respective wards, the majority of medical students do not exercise or consume healthier foods due to a shortage of time. Another reason is the continuous stress of being a medical student and the burden of their studies would have a negative impact on their diet [13, 14]. However, Medical students must develop healthy eating habits because they will be future healthcare providers. Students and physicians who decide not to live a healthy lifestyle are more reasonably to fail to promote health for their patients in the future [13, 15].

A Few studies have been conducted on the eating habits of medical and non-medical students, but in our study, we engage students from all the health colleges to compare with their colleagues from other colleges. This study aimed to increase nutritional awareness among NBU students and to spread proper nutritional practices in order to enjoy a healthy life. The specific objectives of the study were to identify and compare eating habits between students at

health colleges and non-health colleges at NBU, and to determine the relationship between students' eating habits and their sociodemographic, lifestyle, and psychological factors.

## Methods

### Study design

A comparative cross-section study.

### Setting

This study was conducted on students at NBU from the first of March to the end of April 2023. NBU is located in the Northern Saudi Arabia and comprises 13 colleges located in three cities namely Arar, Rafha, and Turaif. These colleges were divided into four health colleges (College of Medicine, College of Medical Sciences, College of Pharmacy, and College of Nursing) and nine non-health colleges (College of Sciences, College of Sciences and Arts in Rafha, College of Engineering, College of Sciences and Arts in Turaif, College of Computing and Information Technology, College of Education and Arts, College of Family and Consumer Sciences, College of Business Administration, and Applied College in Arar).

### Study population and sampling strategy

The estimated sample size to compare proportions between two samples was calculated using STATA/SE version 11.2 for Windows (STATA Corporation, College Station, Texas). Two samples of 240 from health colleges and non-health colleges were selected, assuming a 50% prevalence of the outcome, a 15% difference between health and non-health colleges, a 95% confidence interval, a 5% margin of error, and a power of 90%.

A convenient sample was selected from health colleges and non-health colleges at NBU. All students at NBU were candidates for the study, including both male and female students. Students with chronic conditions such as diabetes mellitus, depression, or eating disorders, those taking long-term medications that affect body weight such as estrogen or corticosteroids, those who underwent bariatric surgery, pregnant, or athletes were excluded from the study.

### Data collection

A self-administered online questionnaire using Google Forms was used to collect data. The questionnaire contains an introductory paragraph that informs participants of the study's aims, the confidentiality of their responses, and the freedom to accept or decline participation in the study.

A predesigned and validated questionnaire that was employed in previous studies in Saudi Arabia was used to collect data on sociodemographic characteristics, eating habits, and eating-related psychological factors [16–18]. The questionnaire comprised three parts: First: Sociodemographic characteristics of the study participants including gender, age, nationality, college, academic years, marital status, tobacco smoking, physical activity, place of residence, living arrangement, household income, number of family members, parents' education, whether parents were separated or not, and whether students' mothers were working or not. Students were asked about their body weight and height to calculate the Body Mass Index (BMI) = weight in kilogram (kg)/ squared height in meters ($m^2$). Second: Eating habits questionnaire that inquired data on having regular meals and ten items regarding the frequency of consumption of different types of food including daily breakfast, number of meals per day, frequency of having snacks per week, frequency of vegetables and fruits consumption per week, weekly consumption of fried food and fast food, frequency of eating with family/friends, amount of water

intake per day, and the type of food consumed. Correct responses to these items were coded as "1" and non-healthy responses were coded as "0". The total dietary score equals the sum of all responses and ranges from zero to 10 [14]. Third: Six eating-related psychological factors [19, 20]. These include eating due to loneliness, being upset or nervous, feeling bored, feeling happy, loss of control when it comes to food, and eating too much until the stomach hurts. Responses to these items were either "rarely", "sometimes', or "often".

Questions about chronic illnesses, medications taken, prior bariatric surgery, pregnancy, and being athletes were included to exclude non-illegible students.

### Ethical consideration

This study was approved by the Local Committee of Bioethics (HAP-09-A-043) at NBU no. (27/44/H). Informed consent was obtained from all students prior to their participation.

### Data analysis

The collected data were entered, presented, and analyzed using the computerized statistical package STATA/SE version 11.2 for Windows (STATA Corporation, College Station, Texas), and MS Excel. Data were summarized using mean ± Standard Deviation (SD), minimum and maximum values for numerical data, and frequency and percentage for qualitative data. Categorical data were compared using the Chi-square test ($x^2$) and the Fisher Exact Test (FET) as appropriate. The Mann-Whiteny test (z) was used to compare numerical data that was found not normally distributed using the Shapiro-Wilk W test. The median dietary score was used to dichotomize dietary scores and logistic regression of adequate dietary habits (score≥4) conditioned on potential sociodemographic and psychological factors was carried out to identify significant predictors. Statistical significance was considered at P<0.05.

### Results

The study recruited 480 eligible Saudi students equally from health and non-health colleges. Table 1 shows sociodemographic characteristics for the studied students. Regarding the students from health colleges, 63.7% were females, with an average age of 21.3(±1.8). In the non-health colleges, most of the participants were females 69.6% and their mean age was 21.0 (±1.7). More than half of the students from health colleges were from clinical years (fourth academic year—internship) while in the non-health colleges, 63.75% of students were from first to third academic years (P<0.001). Most of the participants were single, living in owned homes with their families, had household monthly income between 3000–10000 SR. More than half of the participants had 6–9 family members and less than 10% had separated parents and were smokers.

Students at health colleges were more likely to have highly educated parents (P = 0.001) and working mothers (P<0.001) than those of non-health colleges. No regular physical activity was reported by 45.8% and 40.4% of students of health and non-health colleges, respectively (P = 0.03). There were no significant differences in BMI between students of health and non-health colleges.

Fig 1 shows the frequency distribution of eating habits reported by the study participants. Only 37.1% and 39.2% of students from health and non-health colleges, respectively, reported having regular meals. Less than half of the participants ate breakfast daily and received three or more daily meals with three or more snacks per week. Most students ate vegetables and legumes less than three times a week (60.4% and 62.9% for students of health and non-health colleges, respectively) as well as fruits (85.0% and 83.7% for students of health and non-health colleges, respectively). Around 57.1% of students from health colleges and 63.3% of students

**Table 1. Sociodemographic characteristics of students at health and non-health colleges, NBU.**

| Characteristic | | | Health colleges (N. = 240) | | Non-health colleges (N. = 240) | | $X^2$ | P |
|---|---|---|---|---|---|---|---|---|
| | | | N. | % | N. | % | | |
| Gender | | Female | 153 | 63.7 | 167 | 69.6 | 1.84 | 0.17 |
| | | Male | 87 | 36.2 | 73 | 30.4 | | |
| Age (year) | | <21 | 88 | 36.7 | 108 | 45.0 | 3.49 | 0.17 |
| | | 21–23 | 102 | 42.5 | 87 | 36.2 | | |
| | | ≥23 | 50 | 20.8 | 45 | 18.7 | | |
| Academic year | | First | 33 | 13.7 | 36 | 15.0 | 46.74 | <0.001 |
| | | Second | 30 | 12.5 | 57 | 23.7 | | |
| | | Third | 51 | 21.2 | 60 | 25.0 | | |
| | | Fourth | 63 | 26.2 | 70 | 29.2 | | |
| | | Fifth | 29 | 12.1 | 17 | 7.0 | | |
| | | Sixth | 15 | 6.2 | 0 | 0.0 | | |
| | | Internship | 19 | 7.9 | 0 | 0.0 | | |
| Marital status | | Single | 231 | 96.2 | 220 | 91.7 | FET | 0.06 |
| | | Married | 9 | 3.7 | 18 | 7.5 | | |
| | | Divorced | 0 | 0.0 | 2 | 0.8 | | |
| Place of residence | | Owned | 205 | 85.4 | 197 | 82.1 | 0.98 | 0.32 |
| | | Rented | 35 | 14.6 | 43 | 17.9 | | |
| Living arrangement | | Live alone | 23 | 9.6 | 12 | 5.0 | 3.73 | 0.05 |
| | | Live with family | 217 | 90.4 | 228 | 95.0 | | |
| Household monthly income (SR) | | 3,000–10,000 | 207 | 86.2 | 213 | 88.7 | 0.68 | 0.41 |
| | | >10,000 | 33 | 13.7 | 27 | 11.2 | | |
| Number of family members | | ≤5 | 38 | 15.8 | 35 | 14.6 | 0.16 | 0.92 |
| | | 6–9 | 138 | 57.5 | 139 | 57.9 | | |
| | | ≥10 | 64 | 26.7 | 66 | 27.5 | | |
| Mothers' educational level | | Not educated | 18 | 7.5 | 20 | 8.3 | 16.14 | 0.001 |
| | | Low | 30 | 12.5 | 45 | 18.7 | | |
| | | Moderate | 50 | 20.8 | 76 | 31.7 | | |
| | | High | 142 | 59.2 | 99 | 41.2 | | |
| Fathers' educational level | | Not educated | 11 | 4.6 | 21 | 8.7 | 16.35 | 0.001 |
| | | Low | 20 | 8.3 | 24 | 10.0 | | |
| | | Moderate | 72 | 30.0 | 101 | 42.1 | | |
| | | High | 137 | 57.1 | 94 | 39.2 | | |
| Separated parents | | No | 223 | 92.9 | 221 | 92.1 | 0.12 | 0.73 |
| | | Yes | 17 | 7.1 | 19 | 7.9 | | |
| Working mother | | No | 117 | 48.7 | 168 | 70.0 | 22.46 | <0.001 |
| | | Yes | 123 | 51.2 | 72 | 30.0 | | |
| Smoking | | No | 222 | 92.5 | 220 | 91.7 | 0.11 | 0.73 |
| | | Yes | 18 | 7.5 | 20 | 8.3 | | |
| Regular physical activity | | No | 110 | 45.8 | 97 | 40.4 | 9.28 | 0.03 |
| | | One day | 17 | 7.1 | 26 | 10.8 | | |
| | | Two days | 48 | 20.0 | 31 | 12.9 | | |
| | | More than two days | 65 | 27.1 | 86 | 35.8 | | |

(*Continued*)

**Table 1.** (Continued)

| Characteristic | | Health colleges (N. = 240) | | Non-health colleges (N. = 240) | | X² | P |
|---|---|---|---|---|---|---|---|
| | | N. | % | N. | % | | |
| BMI (kg/m²) | Underweight (<18.5) | 35 | 14.6 | 36 | 15.0 | 3.31 | 0.35 |
| | Normal weight (18.5–24.9) | 120 | 50.0 | 133 | 55.4 | | |
| | Overweight (25–29.9) | 58 | 24.2 | 42 | 17.5 | | |
| | Obese (≥30) | 27 | 11.2 | 29 | 12.1 | | |

X²:Chi-square test; FET: Fisher Exact Test; z: the Mann Whitney test; statistical significance was considered at P<0.05; BMI: Body Mass Index; SR: Saudi Rial; NBU: Northern Border University

from non-health colleges students ate fried foods less than three times a week. Only 5% of participants reported that they rarely eat fast food. About 50.4% and 47.1% of students at health and non-health colleges, respectively ate with their family and friends daily. Most of the students reported having less than two liters of water per day (71.7% and 73.7% for students of health and non-health colleges, respectively). More than half of students (54.2% and 59.2% for students of health and non-health colleges, respectively) ate a variety of food in balance. There

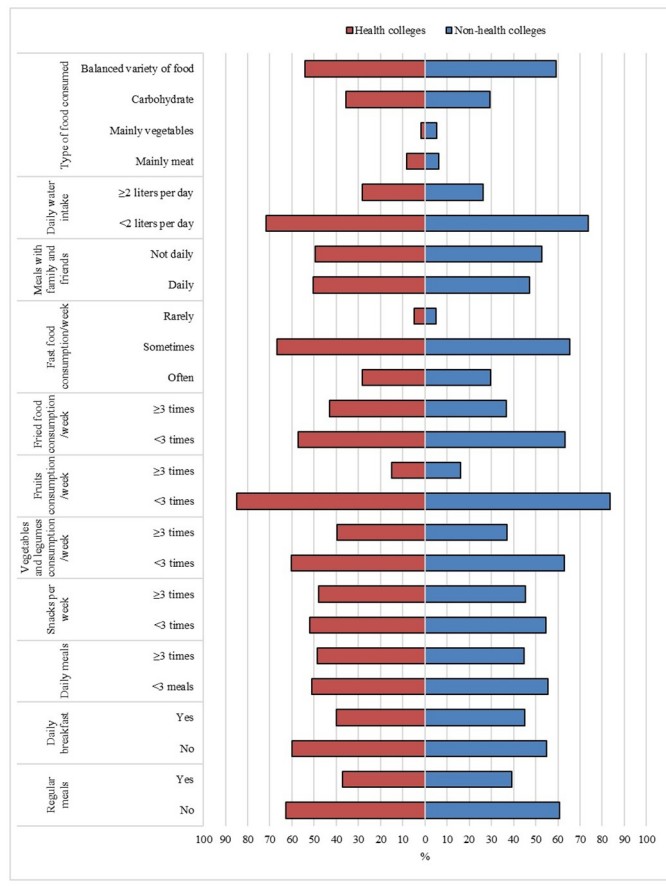

**Fig 1. Eating habits among students at health and non-health colleges, Northern Border University.**

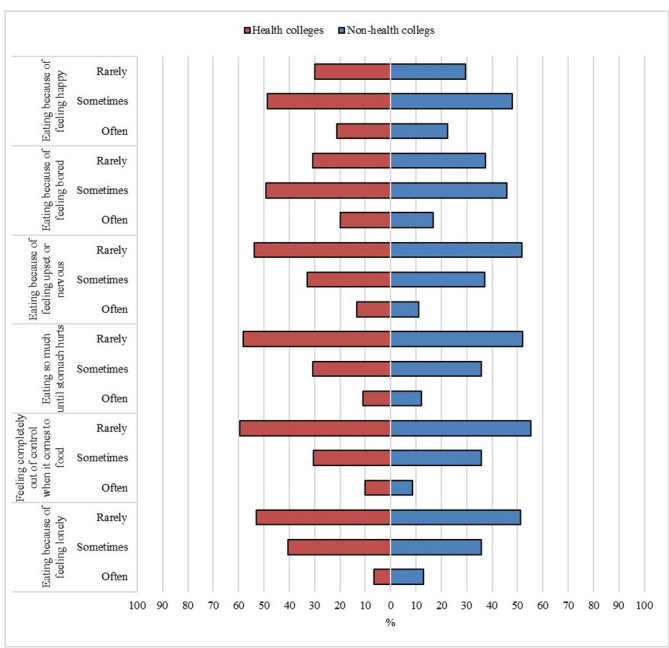

**Fig 2. Psychological factors affecting eating habits among students at health and non-health colleges, Northern Border University.**

were no significant differences in eating habits between student of health and non-health colleges.

Psychological factors affecting eating habits among studied students are demonstrated in Fig 2. More than half of the students of health and non-health colleges reported that they rarely ate because they felt lonely (52.9% and 51.2%), felt completely out of control about food (59.6% and 55.4%), ate so much that their stomach hurts (58.3% and 52.1%), and ate because of feeling upset (53.7% and 51.7%). While 49.2% and 45.8% of students from health and non-health colleges, respectively, reported that they sometimes ate because they felt bored and 48.7% and 47.9%, respectively, sometimes ate when they felt happy. These differences were statistically non-significant.

A total of 133 students (55.4%) from health colleges had adequate eating habits (Table 2). These were more likely in students aged 21–23 years (P = 0.02) and those living with their families (P = 0.04). Parental separation and physical inactivity were associated with inadequate eating habits at P = 0.02 and P = 0.047, respectively. More than half of students (55.6%) with adequate eating habits were of normal weight (P = 0.03).

Table 3 shows that a total of 141 students (58.7%) from non-health colleges had adequate eating habits. Adequate eating habits were less likely among students with separated parents (P = 0.01) and those who practice sports (P = 0.001).

Table 4 shows the relationship between psychological factors and eating habits among students of health and non-health colleges. For students from non-health colleges, adequate eating habits were more frequent in students who rarely felt completely out of control about food (P = 0.007) and ate so much that their stomach hurts (P<0.001). About 56% of students who rarely ate because of feeling upset or nervous had adequate eating habits (P = 0.08). Psychological factors had non-significant differences in eating habits among students at health colleges.

A logistic regression model was constructed to determine the Odds Ratio and 95% Confidence Intervale (OR and 95% CI) for adequate eating habits conditioned on significant

**Table 2. Relationship between eating habits and sociodemographic characteristics of students at health colleges, NBU.**

| Characteristics | | Inadequate eating habits (N. = 107) | | Adequate eating habits (N. = 133) | | X² | P |
|---|---|---|---|---|---|---|---|
| | | N. | % | N. | % | | |
| Gender | Female | 72 | 67.3 | 81 | 60.9 | 1.05 | 0.31 |
| | Male | 35 | 32.7 | 52 | 39.1 | | |
| Age (year) | <21 | 49 | 45.8 | 39 | 29.3 | 7.37 | 0.02 |
| | 21–23 | 37 | 34.6 | 65 | 48.9 | | |
| | ≥23 | 21 | 19.6 | 29 | 21.8 | | |
| Academic year | First | 19 | 17.8 | 14 | 10.5 | 8.40 | 0.21 |
| | Second | 17 | 15.9 | 13 | 9.8 | | |
| | Third | 19 | 17.8 | 32 | 24.1 | | |
| | Fourth | 25 | 23.4 | 38 | 28.6 | | |
| | Fifth | 11 | 10.3 | 18 | 13.5 | | |
| | Sixth | 5 | 4.8 | 10 | 7.5 | | |
| | Internship | 11 | 10.3 | 8 | 6.0 | | |
| Marital status | Single | 103 | 96.3 | 128 | 96.2 | FET | 1.00 |
| | Married | 4 | 3.7 | 5 | 3.8 | | |
| Place of residence | Owned | 88 | 82.2 | 117 | 88.0 | 1.56 | 0.21 |
| | Rented | 19 | 17.8 | 16 | 12.0 | | |
| Living arrangement | Live alone | 15 | 14.0 | 8 | 6.0 | 4.38 | 0.04 |
| | Live with family | 92 | 86.0 | 125 | 94.0 | | |
| Household monthly income (SR) | 3,000–10,000 | 94 | 87.8 | 113 | 85.0 | 0.42 | 0.52 |
| | >10,000 | 13 | 12.1 | 20 | 15.0 | | |
| Number of family members | ≤5 | 21 | 19.6 | 17 | 12.8 | 3.39 | 0.18 |
| | 6–9 | 55 | 51.4 | 83 | 62.4 | | |
| | ≥10 | 31 | 29.0 | 33 | 24.8 | | |
| Mothers' educational level | Not educated | 9 | 8.4 | 9 | 6.8 | 3.76 | 0.29 |
| | Low | 18 | 16.8 | 12 | 9.0 | | |
| | Moderate | 21 | 19.6 | 29 | 21.8 | | |
| | High | 59 | 55.1 | 83 | 62.4 | | |
| Fathers' educational level | Not educated | 5 | 4.7 | 6 | 4.5 | FET | 0.86 |
| | Low | 8 | 7.5 | 12 | 9.0 | | |
| | Moderate | 35 | 32.7 | 37 | 27.8 | | |
| | High | 59 | 55.1 | 78 | 58.6 | | |
| Separated parents | No | 95 | 88.8 | 128 | 96.2 | 5.01 | 0.02 |
| | Yes | 12 | 11.2 | 5 | 3.8 | | |
| Working mother | No | 58 | 54.2 | 59 | 44.4 | 2.30 | 0.13 |
| | Yes | 49 | 45.8 | 74 | 55.6 | | |
| Smoking | No | 99 | 92.5 | 123 | 92.5 | 0.00 | 0.99 |
| | Yes | 8 | 7.5 | 10 | 7.5 | | |
| Playing sports/week | No | 59 | 55.1 | 51 | 38.3 | 7.94 | 0.047 |
| | One day | 6 | 5.6 | 11 | 8.3 | | |
| | Two days | 15 | 14.0 | 33 | 24.8 | | |
| | > two days | 27 | 25.2 | 38 | 28.6 | | |
| BMI (kg/m²) | Underweight (<18.5) | 23 | 21.5 | 12 | 9.0 | 9.16 | 0.03 |
| | Normal weight (18.5–24.9) | 46 | 43.0 | 74 | 55.6 | | |
| | Overweight (25–29.9) | 28 | 26.2 | 30 | 22.6 | | |
| | Obese (≥30) | 10 | 9.3 | 17 | 12.8 | | |

X²:Chi-square test; FET: Fisher Exact Test; statistical significance was considered at P<0.05; BMI: Body Mass Index; SR: Saudi Rial, NBU: Northern Border University

**Table 3. Relationship between eating habits and sociodemographic characteristics of students at non-health colleges, NBU.**

| Characteristics | | | Inadequate eating habits (N. = 99) | | Adequate eating habits (N. = 141) | | $X^2$ | P |
|---|---|---|---|---|---|---|---|---|
| | | | N. | % | N. | % | | |
| Gender | | Female | 66 | 66.7 | 101 | 71.6 | 0.68 | 0.41 |
| | | Male | 33 | 33.3 | 40 | 28.4 | | |
| Age (year) | | <21 | 49 | 49.5 | 59 | 41.8 | 4.79 | 0.09 |
| | | 21–23 | 28 | 28.3 | 59 | 41.8 | | |
| | | ≥23 | 22 | 22.2 | 23 | 16.3 | | |
| Academic year | | First | 15 | 15.1 | 21 | 14.9 | 4.54 | 0.34 |
| | | Second | 26 | 26.3 | 31 | 22.0 | | |
| | | Third | 27 | 27.3 | 33 | 23.4 | | |
| | | Fourth | 22 | 22.2 | 48 | 34.0 | | |
| | | Fifth | 9 | 9.1 | 8 | 5.7 | | |
| Marital status | | Single | 92 | 92.9 | 128 | 90.8 | FET | 0.81 |
| | | Married | 6 | 6.1 | 12 | 8.5 | | |
| | | Divorced | 1 | 1.0 | 1 | 0.7 | | |
| Place of residence | | Owned | 80 | 80.8 | 117 | 83.0 | 0.19 | 0.67 |
| | | Rented | 19 | 19.2 | 24 | 17.0 | | |
| Living arrangement | | Live alone | 6 | 6.1 | 6 | 4.3 | 0.40 | 0.53 |
| | | Live with family | 93 | 93.9 | 135 | 95.7 | | |
| Household monthly income (SR) | | 3,000–10,000 | 91 | 91.9 | 122 | 86.5 | 1.69 | 0.19 |
| | | >10,000 | 8 | 8.1 | 19 | 13.5 | | |
| Number of family members | | ≤5 | 14 | 14.1 | 21 | 14.9 | 0.04 | 0.98 |
| | | 6–9 | 58 | 58.6 | 81 | 57.4 | | |
| | | ≥10 | 27 | 27.3 | 39 | 27.7 | | |
| Mothers' educational level | | Not educated | 7 | 7.1 | 13 | 9.2 | 2.56 | 0.46 |
| | | Low | 21 | 21.2 | 24 | 17.0 | | |
| | | Moderate | 35 | 35.3 | 41 | 29.1 | | |
| | | High | 36 | 36.4 | 63 | 44.7 | | |
| Fathers' educational level | | Not educated | 8 | 8.1 | 13 | 9.2 | 3.51 | 0.32 |
| | | Low | 7 | 7.1 | 17 | 12.1 | | |
| | | Moderate | 48 | 48.5 | 53 | 37.6 | | |
| | | High | 36 | 36.4 | 58 | 41.1 | | |
| Separated parents | | No | 86 | 86.9 | 135 | 95.7 | 6.28 | 0.01 |
| | | Yes | 13 | 13.1 | 6 | 4.3 | | |
| Working mother | | No | 73 | 73.7 | 95 | 67.4 | 1.12 | 0.29 |
| | | Yes | 26 | 26.3 | 46 | 32.6 | | |
| Smoking | | No | 88 | 88.9 | 132 | 93.6 | 1.70 | 0.19 |
| | | Yes | 11 | 11.1 | 9 | 6.4 | | |
| Playing sports/week | | No | 54 | 54.5 | 43 | 30.5 | 16.23 | 0.001 |
| | | One day | 11 | 11.1 | 15 | 10.6 | | |
| | | Two days | 11 | 11.1 | 20 | 14.2 | | |
| | | > two days | 23 | 23.2 | 63 | 44.7 | | |
| BMI (kg/m$^2$) | | Underweight (<18.5) | 17 | 17.2 | 19 | 13.5 | 4.36 | 0.22 |
| | | Normal weight (18.5–24.9) | 47 | 47.5 | 86 | 61.0 | | |
| | | Overweight (25–29.9) | 21 | 21.2 | 21 | 14.9 | | |
| | | Obese (≥30) | 14 | 14.1 | 15 | 10.6 | | |

$X^2$:Chi-square test; FET: Fisher Exact Test; statistical significance was considered at P<0.05; BMI: Body Mass Index; SR: Saudi Rial; NBU: Northern Border University

**Table 4. Relationship between psychological factors and eating habits among students of health and non-health colleges, NBU.**

| Psychological factors | | Health colleges | | | | X² | P | Non-health colleges | | | | X² | P |
|---|---|---|---|---|---|---|---|---|---|---|---|---|---|
| | | Inadequate eating habits (N. = 107) | | Adequate eating habits (N. = 133) | | | | Inadequate eating habits (N. = 99) | | Adequate eating habits (N. = 141) | | | |
| | | N. | % | N. | % | | | N. | % | N. | % | | |
| Eating because of feeling lonely | Often | 9 | 8.4 | 7 | 5.3 | 1.10 | 0.58 | 15 | 15.1 | 16 | 11.3 | 0.92 | 0.63 |
| | Sometimes | 41 | 38.3 | 56 | 42.1 | | | 36 | 36.4 | 50 | 35.5 | | |
| | Rarely | 57 | 53.3 | 70 | 52.6 | | | 48 | 48.5 | 75 | 53.2 | | |
| feeling completely out of control when it comes to food | Often | 11 | 10.3 | 13 | 9.8 | 0.22 | 0.90 | 11 | 11.1 | 10 | 7.1 | 9.79 | 0.007 |
| | Sometimes | 34 | 31.8 | 39 | 29.3 | | | 45 | 45.4 | 41 | 29.1 | | |
| | Rarely | 62 | 57.9 | 81 | 60.9 | | | 43 | 43.4 | 90 | 63.8 | | |
| Eating so much until stomach hurts | Often | 14 | 13.1 | 12 | 9.0 | 2.97 | 0.23 | 22 | 22.2 | 7 | 5.0 | 22.32 | <0.001 |
| | Sometimes | 37 | 34.6 | 37 | 27.8 | | | 40 | 40.4 | 46 | 32.6 | | |
| | Rarely | 56 | 52.3 | 84 | 63.2 | | | 37 | 37.4 | 88 | 62.4 | | |
| Eating because of feeling upset or nervous | Often | 16 | 14.9 | 16 | 12.0 | 0.64 | 0.73 | 16 | 16.2 | 11 | 7.8 | 4.95 | 0.08 |
| | Sometimes | 33 | 30.8 | 46 | 34.6 | | | 38 | 38.4 | 51 | 36.2 | | |
| | Rarely | 58 | 54.2 | 71 | 53.4 | | | 45 | 45.4 | 79 | 56.0 | | |
| Eating because of feeling bored | Often | 23 | 21.5 | 25 | 18.8 | 0.44 | 0.80 | 20 | 20.2 | 20 | 14.2 | 3.24 | 0.20 |
| | Sometimes | 53 | 49.5 | 65 | 48.9 | | | 48 | 48.5 | 62 | 44.0 | | |
| | Rarely | 31 | 29.0 | 43 | 32.3 | | | 31 | 31.3 | 59 | 41.8 | | |
| Eating because of feeling happy | Often | 19 | 17.8 | 32 | 24.1 | 1.43 | 0.49 | 18 | 18.2 | 36 | 25.5 | 2.15 | 0.34 |
| | Sometimes | 55 | 51.4 | 62 | 46.6 | | | 52 | 52.5 | 63 | 44.7 | | |
| | Rarely | 33 | 30.8 | 39 | 29.3 | | | 29 | 29.3 | 42 | 29.8 | | |

$X^2$:Chi-square test; Statistical significance was considered at P<0.05; NBU: Northern Border University

predictors (Table 5). For students of health colleges, adequate eating habits were more likely among elder students aged 21–23 years (OR = 2.58 and 95% CI = 1.33–4.99; P = 0.005), and those who lived with their families (OR = 3.04 and 95%CI = 1.10–8.41; P = 0.03). While students with separated parents were less likely to have healthy eating habits (OR = 0.23 and 95% CI = 0.07–0.75; P = 0.01). Regular physical activity was associated with appropriate eating habits (P<0.05). Normal-weight students were more likely to eat properly than underweight students (OR = 2.87 and 95%CI = 1.23–6.69; P = 0.01).

For students at non-health colleges, adequate eating habits were less likely among students with separated parents (OR = 0.19 and 95%CI = 0.06–0.60; P = 0.005) and those who reported that they sometimes eat because of feeling happy (OR = 0.40 and 95%CI = 0.18–0.87; P = 0.02). Exercising regularly for more than two days per week was associated with an increased likelihood of healthy eating (OR = 3.61 and 95%CI = 1.81–7.20; P<0.001). Students who rarely or sometimes ate so much until their stomach hurts were more likely to eat properly compared to those who often did this (OR = 9.19 and 95%CI = 3.21–26.37; P<0.001 and OR = 4.73 and 95%CI = 1.63–13.69; P = 0.004, respectively).

## Discussion

This study revealed that 44.6% and 41.3% of students at health and non-health colleges, respectively, had inadequate eating habits (dietary habits scores<4), with an average eating habit score of 3.9. Most students had irregular meals, skipped breakfast, and consumed insufficient vegetables, fruits, and water. In addition, a considerable proportion of students frequently consumed fried and fast food. Similar results were reported among university students in Saudi

**Table 5. Logistic regression of adequate eating habits conditioned on significant risk factors among students of health and non-health colleges, NBU.**

| Risk factors | Health colleges (N. = 240) | | Risk factors | Non-health colleges (N. = 240) | |
|---|---|---|---|---|---|
| | OR (95% CI) | P | | OR (95% CI) | P |
| **Separated parents** | | | **Separated parents** | | |
| Yes vs. no | 0.23(0.07–0.75) | 0.01 | Yes vs. no | 0.19(0.06–0.60) | 0.005 |
| **Playing sports/week** | | | **Playing sports/week** | | |
| No | 1.00 | | No | 1.00 | |
| One day | 4.22(1.22–14.63) | 0.02 | One day | 2.20(0.84–5.78) | 0.11 |
| Two days | 3.40(1.54–7.48) | 0.002 | Two days | 1.82(0.75–4.40) | 0.18 |
| > Two days | 2.23(1.12–4.44) | 0.02 | > Two days | 3.61(1.81–7.20) | <0.001 |
| **BMI (kg/m$^2$)** | | | **Eating so much until stomach hurts** | | |
| Underweight (<18.5) | 1.00 | | | | |
| Normal weight (18.5–24.9) | 2.87(1.23–6.69) | 0.01 | Often | 1.00 | |
| Overweight (25–29.9) | 1.86(0.72–4.77) | 0.20 | Sometimes | 4.73(1.63–13.69) | 0.004 |
| Obese (≥30) | 3.95(1.28–12.20) | 0.02 | Rarely | 9.19(3.21–26.37) | <0.001 |
| **Age (year)** | | | **Eating because of feeling happy** | | |
| <21 | 1.00 | | Often | 1.00 | |
| 21–23 | 2.58(1.33–4.99) | 0.005 | Sometimes | 0.40(0.18–0.87) | 0.02 |
| ≥23 | 2.02(0.92–4.45) | 0.08 | Rarely | 0.59(0.25–1.38) | 0.22 |
| **Living arrangement** | | | | | |
| Live with family vs. live alone | 3.04(1.10–8.41) | 0.03 | | | |

OR: Odd's Ratio; 95% CI: 95% Confidence Interval; statistical significance was considered at P<0.05; NBU: Northern Border University.

Arabia. Most medical students at King Abdulaziz University, Jeddah had unhealthy dietary habits including irregular meals, inadequate intake of fruits and vegetables, and frequent eating of fried and fast foods with an average eating habit score of 4.2 [18]. Again, the majority of medical students from five Saudi universities had inadequate eating habits [16]. Moreover, studies in Saudi Arabia to assess eating habits in university students using the Eating Attitude Test (EAT-26) revealed prevalence of high-risk dietary disorders ranging from 29.4% to 35.4% [21–23]. Also, a study in the United Arab Emirates [24] showed poor eating habits among university students such as not meeting the recommended intake of vegetables, fruits, and water [20]. Outside the Gulf region, poor eating habits were reported by most university students in Malaysia [14] and Darussalam [25]. While 22.75% of medical students in Pakistan [26] and 17% in Lebanon [27] were at high risk of eating disorders. The differences in eating habits can be due to different cultural and environmental factors besides the variable numbers of students involved.

Our results showed no differences in BMI between students of health and non-health colleges with 35.4% and 29.7% of students from health and non-health colleges were overweight/obese, respectively. The prevalence rates of overweight/obesity in recent studies on Saudi university students were 31% [21] and 34.7% [28]. This was comparable to findings in the UAE [20, 24]. The prevalence of overweight/obesity in university students in other Arab countries were 36% in Jordan [29] and 35.9% in Upper Egypt [30]. While smaller proportion of university students were overweight/obese in Iran (21.3%) [31] and Malaysia (24.44%) [32]. Obesity has been linked to poor dietary habits [8].

In the present study, only 27.1% and 35.8% of students at health and non-health colleges practiced sports more than two days per week while 45.8% and 40.4% of students, respectively from health and non-health colleges reported not playing sports. These findings were

consistent in Saudi Arabia [18, 33]. Al-Nakeeb et al. reported lower level of physical activity among young Saudi people (26%) compared to adolescents in two British cities (72.3–77.2%) [33]. In the UAE, 32.7–39% of university students did not practice sports [20, 34]. The prevalence of physical activity in the countries of the Gulf Cooperation Council (GCC) is generally low with men more active than women due to socio-cultural and environmental factors of the Gulf region, which predispose to many non-communicable diseases including obesity [35].

Our study revealed that only 37.1% and 39.2% of students at health and non-health colleges, respectively reported eating regular meals, which was comparable to previous studies in Qassim (36.7%) [36] and in Abha (31%) [17] while was lower than that reported in Jeddah (50.5%) [18]. Studies outside Saudi Arabia reported higher prevalence of regular meals in university students such as in Malaysia (57.6%) [14], Lebanon (61.4%) [37], Darussalam (74.6%) [25], and China (83.6%) [38].

In this study, 40% and 45% of students at the health and non-health colleges ate breakfast daily. Comparable results were reported among university students in the UAE (45.4%) [20], Darussalam (42.6%) [25], and Malaysia (43.9%) [14]. While 41% and 68.7% of anemic and non-anemic students in Bangladesh had breakfast daily [39]. However, smaller proportions were reported among Saudi medical students (34.7%) [18] and university students in Lebanon (31.8%) [37]. While higher proportion (59.7%) of university students in coastal Kenya had daily breakfast [40]. Skipping breakfast can be due to lack of time, reduced appetite and not feeling hungry early morning, and can negatively affect the academic performance of students and their other eating habits e.g., frequent snacks [41, 42].

About half of the studied students had snacks three or more times per week. Consumption of fried food was reported by 42.9% and 36.7% of students of health and non-health colleges at least three times per week, and only 5% rarely ate fast food. While 60.4% and 47.3% of Saudi medical students in Jeddah reported having snacks and eating fried foods at least three times a week, respectively, and 6.1% never eating fast food [18]. In Abha, 35.96% of university male students reported eating snacks daily [17]. In addition, 88.7% of Saudi male students reported eating fast food at least one time per week [43]. Meanwhile, 42.4% of Malaysian students had snacks three or more times per week [14]. While 82.2% of students at Universiti Brunei Darussalam reported eating snacks between meals and 60.7% ate fried food 3–5 times per week [25]. In Kenya, 47.2% of university students reported eating fried food 3 or more times per week and only 15.3% ate with snacks on daily basis [40]. Fast food is convenient, easily accessible, and often fits students' busy schedules. Additionally, peer influence plays a significant role as eating at fast-food chains is a social activity among young people besides the high frequency of skipping breakfast among studied students. Also, fast food is marketed heavily towards teens and young people.

The WHO recommended daily consumption of five servings of fruits and vegetables that is equivalent to 400 grams per day [44]. Also, the Saudi Ministry of Health recommended consumption of 3–5 servings of vegetables and 2–4 servings of fruits per day for healthy food and protection against many diseases [45]. However, most students in the present study reported eating vegetables and fruits less than three times a week. Previous studies in Saudi Arabia reported similar results [17, 18, 46]. Correspondingly, only 28.7% and 34% of university students in the UAE ate fruits and vegetables daily [20] and 30.5% of Lebanese university students ate colored vegetables daily [37]. Meanwhile 42.9% and 41.9% of university students in Brunei ate vegetables and fruits 3–5 times per week [25] and most Kenyan students reported eating vegetables (87.5%) and fruits (69.4%) daily [40].

In this study, most students drank less than the recommended daily water intake (two liters and two and half liters for females and males, respectively according to the European Food Safety Authority) [47]. In agreement, 90% of Saudi medical students reported drinking less

than two liters of water per day [18]. Similarly, more than four fifths of UAE university students consumed less than two liters per day [20]. In Iran, the average amount of daily fluid intake by university students was 1.7 liters [48]. Outside the Gulf area, 57.8% of university students reported drinking at least two liters of water per day [25].

About half of the studied students reported eating with their families and friends daily and more than half ate a variety of food in balance. These results agreed with previous studies in Saudi Arabia [17, 18]. However, a smaller proportion of students reported eating with their families and friends in Lebanon (42.7%) [37]. Meanwhile most Malaysian students (81.8%) ate with their families [14].

This study revealed that most students rarely ate because of feeling lonely or nervous, they rarely felt completely out of control about food, and rarely ate so much that their stomachs hurt. While about one fifth reported they often ate because of boredom or happiness. These findings agree with previous records of Saudi medical students in Jeddah where only 10–21% of students reported that they often eat due to psychological factors [18]. In the UAE, more than 80% of the University of Sharjah students ate out of boredom or happiness, and more than half ate out of sadness [20]. The association between psychological factors and eating habits has been proved. Moynihan et al. reported that boredom increased caloric consumption to distract from this feeling [49]. In addition, food intake was significantly increased in the positive mood condition compared to the neutral condition while the negative condition had no effect on food consumption. Moreover, there was a positive correlation between the total calories consumed and improved mood [50]. A meta-analysis revealed that induced positive and negative mood conditions were associated with increased eating in both healthy and patients with eating disorders [51]. Mood changes associated with changes in food choices and quantities. The link between mood and food can be mediated by hormones such as ghrelin and leptin and neurotransmitters such as dopamine and serotonin [52].

In this study, 55.4% and 58.75% of students of health and non-health colleges reported sound eating habits. Healthy eating was more likely in students at health colleges who had normal body weight, aged 21–23 years, and those living with their families. While parental separation and physical inactivity were associated with inadequate eating habits among students of both health and non-health colleges. Similarly, higher dietary scores were more likely among Saudi medical students who lived with their families, whose parents were not separated, were physically active, never smokers, and lived in an owned place (P<0.05) [18]. In line with these findings, unhealthy feeding was more frequent in unmarried students, those not living with their families, smokers, and physically inactive [20]. It was suggested that lack of family supervision in students living alone predisposed to unhealthy habits such as unhealthy food choices and lack of physical activity [53].

In the present study, psychological factors such as feeling out of control about food, too much eating till stomach hurts, and eating because of being upset or nervous were unlikely among students with adequate eating habits. Similarly, Alzahrani et al. recorded higher eating scores among medical students who never ate out of feeling lonely, felt out of control when it came to food, ate until their stomach hurt, ate because of feeling upset or nervous, or ate because of boredom compared to those who did so often or sometimes (P<0.05) [18]. In Kuwait, stress was associated with consumption of unhealthy food such as fast-food rich in fats, sugars, and soft drinks particularly in female students [54]. In agreement, unhealthy dietary habits and obesity were associated with uncontrollable eating [20, 53].

This comparative cross-sectional study examined dietary habits among students of both health and non-health colleges at the NBU. A wide range of underlying factors for unsatisfactory dietary habits have been studied including sociodemographic characteristics, lifestyle, and psychological factors. The study presented important factors with potential implications for

improving the dietary habits of university students, which could have long-term effects on their health and well-being.

However, our results are based on self-reported data with a risk of bias due to social desirability, over-reporting of healthy habits, or under-reporting of unhealthy habits. The absence of a specified period of recall and the inability to establish causality are also important considerations. Additionally, the fact that factors like chronic diseases and dietary disorders have not been studied highlights a potential area for future research. Conducting larger-scale studies across diverse population subgroups would indeed provide a more comprehensive understanding of eating habits and the underlying factors influencing them.

Finally, unhealthy eating habits were prevalent among students at NBU irrespective of the type of study. This was evident in terms of irregular meals, skipping breakfast, and insufficient intake of fruits, vegetables, and water. Young age, parental separation, lack of physical activity, living away from family, and low BMI were associated with unhealthy eating habits. Some psychological factors, such as overeating until the stomach hurts and eating to feel happy, increase the risk of eating unhealthy food. Given these findings, implementing initiatives to promote nutrition and healthy eating habits within the university environment is crucial. This could involve educational campaigns focused on nutrition, as well as integrating nutritional education into the university curricula. Encouraging and supporting students in adopting healthier diets and lifestyles can have long-lasting benefits for their overall well-being and academic performance.

## Supporting information

**S1 File.**
(DOCX)

**S1 Data.**
(XLSX)

## Acknowledgments

The authors are thankful to all students who agreed to participate in this study.

## Author Contributions

**Conceptualization:** Hanaa E. Bayomy, Shmoukh Mushref Alruwaili, Razan Ibrahim Alsayer, Nuof Khalid Alanazi, Dana Ahmed Albalawi, Khulud Hamed Al Shammari, Mariam Mahmoud Moussa.

**Data curation:** Shmoukh Mushref Alruwaili, Razan Ibrahim Alsayer, Nuof Khalid Alanazi, Dana Ahmed Albalawi, Khulud Hamed Al Shammari.

**Formal analysis:** Hanaa E. Bayomy, Mariam Mahmoud Moussa.

**Methodology:** Hanaa E. Bayomy, Mariam Mahmoud Moussa.

**Software:** Hanaa E. Bayomy.

**Supervision:** Hanaa E. Bayomy.

**Writing – original draft:** Hanaa E. Bayomy, Shmoukh Mushref Alruwaili, Razan Ibrahim Alsayer, Nuof Khalid Alanazi, Dana Ahmed Albalawi, Khulud Hamed Al Shammari, Mariam Mahmoud Moussa.

**Writing – review & editing:** Hanaa E. Bayomy, Shmoukh Mushref Alruwaili, Razan Ibrahim Alsayer, Nuof Khalid Alanazi, Dana Ahmed Albalawi, Khulud Hamed Al Shammari, Mariam Mahmoud Moussa.

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
