## [Decision Letter · Decision Letter 0]

19 Aug 2024

PONE-D-24-23613Eating Habits of Students of Health Colleges and Non-Health Colleges at ‎the Northern Border University in the Kingdom of Saudi ArabiaPLOS ONE

Dear Dr. Bayomy,

Thank you for submitting your manuscript to PLOS ONE. After careful consideration, we feel that it has merit but does not fully meet PLOS ONE’s publication criteria as it currently stands. Therefore, we invite you to submit a revised version of the manuscript that addresses the points raised during the review process.

**Areas for Improvement:**

Authors must work in these areas before we can proceed for this paper. I have major concerns as follows

The aim could be more precisely articulated. For example, how will comparing health and non-health students contribute to the broader understanding of student eating habits? Expanding on this could add more depth to the background.STROBE Checklist is missing and not in the method section? Why was it omitted?**Study Design Limitations**:As a cross-sectional study, causality cannot be established. The manuscript should emphasize this limitation clearly, as it can lead to misinterpretation of associations as causal relationships.**Recall Bias**: The study relies on self-reported data, which may be prone to recall bias. Addressing this limitation and discussing its potential impact on the findings would strengthen the discussion.**Psychological Factors**:The psychological factors mentioned (e.g., overeating until the stomach hurts, eating to feel happy) seem significant but are vaguely described. More detail is needed regarding how these factors were measured. Were standardized scales used, or were these self-reported behaviors? Clarification here would enhance the study's validity.Furthermore, the connection between these psychological factors and unhealthy eating habits could be explored in more depth**Discussion of Results**:The discussion should provide more critical analysis of the results, linking them to existing literature. For example, how do the findings compare to other studies on university students' eating habits, both in Saudi Arabia and internationally?**Conclusion and Recommendations**:The conclusion could be more actionable. While the manuscript calls for initiatives to promote nutrition and healthy eating, it would benefit from specific recommendations, such as examples of successful programs or policies that have worked in other university settings.Additionally, the conclusion should reflect the study's limitations. For example, it should acknowledge the inability to establish causality and the potential influence of unmeasured confounders.**Ethical Considerations**:The manuscript does not mention ethical approval or consent procedures, which are crucial for research involving human subjects. This section should be added, ensuring that the study adheres to ethical guidelines.**Stylistic and Grammatical Issues**:The manuscript could benefit from a thorough proofreading to improve readability. Some sentences are long and complex, which may hinder comprehension. Simplifying these sentences would enhance clarity.

We look forward to receiving your revised manuscript.

Kind regards,

Mohammad Sidiq, PhD Pain Sciences Physiotherapy

Academic Editor

PLOS ONE

Journal Requirements:

2. We note that your Data Availability Statement is currently as follows: “All relevant data are within the manuscript and its Supporting Information files.”

Additional Editor Comments:

While authors have tried to talk about an important issue. I would like them to revise it before it can considered for publication.

Reviewers' comments:

Reviewer's Responses to Questions

**Comments to the Author**

1. Is the manuscript technically sound, and do the data support the conclusions?

Reviewer #1: Partly

Reviewer #2: Yes

2. Has the statistical analysis been performed appropriately and rigorously? 

Reviewer #1: No

Reviewer #2: Yes

3. Have the authors made all data underlying the findings in their manuscript fully available?

Reviewer #1: Yes

Reviewer #2: Yes

4. Is the manuscript presented in an intelligible fashion and written in standard English?

Reviewer #1: Yes

Reviewer #2: Yes

5. Review Comments to the Author

Reviewer #1: Frequent change in the keywords, data from the health college has non uniform academic year distribution, results are partly in line with the objectives, the discussion majorly lack the justification of the fidnings of this study. wherever contradictory findings have been mentioned in the discussion, it lacks the scientific explanation and authors also failed to justify the objective in the discussion.

Revise the Results to apply statistical tools to compare the data between health and non health college students.

Revise the discussion and try to add scientific justification of all your findings.

Reviewer #2: The novelty of the research has been clearly stated. The quality of research is satisfactory. The problem under investigation is of general public interest and of moderate importance. The results have been

clearly analysed and thoroughly discussed with logical explanation. The tabular representation of data is satisfactory. Ethical principles while conducting the research have been appropriately taken care of.

6. PLOS authors have the option to publish the peer review history of their article (what does this mean?). If published, this will include your full peer review and any attached files.

Reviewer #1: No

Reviewer #2: **Yes: **Jaspreet Singh Vij

---

## [Author Response · Author response to Decision Letter 0]

24 Sep 2024

Responses to academic editor and reviewers’ comments

We sincerely appreciate the reviewers for their comments to improve this article. All responses are in red colour.

Areas for improvement :

• STORBE checklist was attached under « Other » files.

• Study objectives were revised.

• Study limitations and potential bias were mentioned.

• Psychological factors were reported subjectively, as self-reported behaviors, and this was mentioned as a limitation of the study.

• Discussion of the results were compared with existing literature in Saudi arabia, other Arab countries and non-Arab countries and finding were justified.

• Conclusion and recommendation : a conclusion based on study findings was provided and recommendations to enhance healthy eating habits of university students through potential activities such as student activities, awareness campaigns, and academic courses were provided.

Ethical considerations : “This study was approved by the Local Committee of Bioethics (HAP-09-A-043) at NBU no. (27/44/H). Informed consent was obtained from all students prior to their participation. “

The data support our findings are attached, given that it does not violate anonymity of participants, and won't partly or collectively personally identify participants.

Response to reviewers’ comments:

1- (page 8) frequent change in key words like eating habits/eating pattern. Can’t understand the reason

Thank you for your comment, “eating patterns” was changed to “eating habits”.

2- (page 9) Use key words what is in your title abstract, avoid changing keywords. Title says eating habits and introduction talk about healthy diet.

Thank you, “A healthy diet is one that provides enough micronutrients…” was changed to “Healthy eating habits provide enough micronutrients….”

3- (page 9) Abbreviation should be immediately after the full name

Thank you, this was done

4- (page 9) Reference 7 does not justify the claim in the introduction.

Thank you, this reference was replaced by “Pan, A., Lin, X., Hemler, E., & Hu, F. B. (2018). Diet and Cardiovascular Disease: Advances and Challenges in Population-Based Studies. Cell metabolism, 27(3), 489–496. https://doi.org/10.1016/j.cmet.2018.02.017”

5- (page 9) Keyword changed

Thank you, “eating behaviors” was changed to “eating habits” 

6- (page 10) Aim changed what has been mentioned in the abstract.

Thank you, in the abstract, “This study aimed to determine and compare eating patterns between students at health and non-health colleges at Northern Border University (NBU), Saudi Arabia and to identify the relationship between eating patterns and sociodemographic and psychological characteristics of students.” was modified to “This study’s objectives were to identify and compare eating habits between students at health and non-health colleges at Northern Border University (NBU), Saudi Arabia and to determine the relationship between students’ eating habits and their sociodemographic, lifestyle, and psychological factors.”

7- (page 10) Keyword changed

Thank you, “eating patterns” was changed to “eating habits”.

8- (page 11) How the response could be wrong? Why the author has used "wrong response" word?

Thank you, “wrong responses” was changed to “non-healthy responses”.

9- (page 12) if all the students of health college has been approached why more than half of the population data is from clinical years while opposite population distribution has been observed in non health college data? Author has already mentioned in the introduction that poor clinical duty related stress is one of the major contributing factor, then more clinical student student in the data may have biased the results.

Thank you for your comment, for health colleges, 47.5% (n.=114) of students were from basic years (years1-3) and 52.5% (n.=126) were from clinical years (years 4-6 and the internship), which is not a significant difference (Z=1.55 and p=0.52), so this difference is unlikely to affect the results.

10- (page 13) Justify how the significance level has been identified, while only change in the percentage has been presented in fig 1.

Thank you, comparisons were carried out using the Chi-square test and statistical significance was considered at p<0.05, as mentioned in the Statistical analysis. “Categorical data were compared using the Chi-square test (x2) and the Fisher Exact Test (FET) as appropriate. ………….. Statistical significance was considered at P<0.05.” Also, here is the table showing details of Figure 1.

Supplementary Table 1: Comparisons of eating habits between students at health and non-health colleges

Eating habits Non-health colleges

(N.=240) Health colleges

(N.=240) X2 P

 N. % N. % 

Do you have regular meals? No 146 60.8 151 62.9 0.22 0.64

 Yes 94 39.2 89 37.1 

Do you have daily breakfast? No 132 55.0 144 60.0 1.23 0.27

 Yes 108 45.0 96 40.0 

Frequency of daily meals Less than three meals 133 55.4 123 51.2 0.84 0.36

 Three or more times 107 44.6 117 48.7 

Frequency of having snacks per week Less than three times 131 54.6 125 52.1 0.30 0.58

 Three or more times 109 45.4 115 47.9 

Weekly consumption of vegetables and legumes Less than three times 151 62.9 145 60.4 0.32 0.57

 Three or more times 89 37.1 95 39.6 

Weekly consumption of fruits Less than three times 201 83.7 204 85.0 0.14 0.71

 Three or more times 39 16.2 36 15.0 

Weekly consumption of fried food Less than three times 152 63.3 137 57.1 1.96 0.16

 Three or more times 88 36.7 103 42.9 

Consumption of fast food Often 71 29.6 68 28.3 0.09 0.95

 Sometimes 157 65.4 160 66.7 

 Rarely 12 5.0 12 5.0 

Meals with family and friends Daily 113 47.1 121 50.4 0.53 0.46

 Not daily 127 52.9 119 49.6 

Water intake Less than two liters per day 177 73.7 172 71.7 0.26 0.61

 Two or more liters per day 63 26.2 68 28.3 

Type of food consumed Mainly meat 15 6.2 20 8.3 7.65 0.05

 Mainly vegetables 13 5.4 4 1.7 

 Carbohydrate 70 29.2 86 35.8 

 Variety of food in balance 142 59.2 130 54.2 

11- (page 13) Justify how the significance level has been identified, while only change in the percentage has been presented in fig 2.

Thank you, comparisons were carried out using the Chi-square test and statistical significance was considered at p<0.05, as mentioned in the Statistical analysis. “Categorical data were compared using the Chi-square test (x2) and the Fisher Exact Test (FET) as appropriate. ………….. Statistical significance was considered at P<0.05.” Also, here is the table showing details of Figure 2.

Supplementary Table 2: Comparisons of eating-related psychological factors between students at health and non-health colleges

Psychological factors Non-health colleges

(N.=240) Health colleges

(N.=240) X2 P

 N. % N. % 

Do you eat because you are feeling lonely? Often 31 12.9 16 6.7 5.51 0.06

 Rarely 123 51.2 127 52.9 

 Sometimes 86 35.8 97 40.4 

Do you feel completely out of control when it comes to food? Often 21 8.7 24 10.0 1.43 0.49

 Rarely 133 55.4 143 59.6 

 Sometimes 86 35.8 73 30.4 

Do you eat so much until stomach hurts? Often 29 12.1 26 10.8 1.91 0.38

 Rarely 125 52.1 140 58.3 

 Sometimes 86 35.8 74 30.8 

Do you eat because of feeling upset or nervous? Often 27 11.2 32 13.3 1.1 0.57

 Rarely 124 51.7 129 53.7 

 Sometimes 89 37.1 79 32.9 

Do you eat because you are feeling bored? Often 40 16.7 48 20.0 2.57 0.28

 Rarely 90 37.5 74 30.8 

 Sometimes 110 45.8 118 49.2 

Do you eat because you are feeling happy? Often 54 22.5 51 21.2 0.11 0.95

 Rarely 71 29.6 72 30.0 

 Sometimes 115 47.9 117 48.7 

12- (page 14) Justification of findings should be written immediately after the claim or fining of this study followed by supporting evidence.

Thank you, inadequate eating habits was defined in the Statistical analysis section as follows “The median dietary score was used to dichotomize dietary scores and logistic regression of adequate dietary habits (score≥4). For more justification “inadequate eating habits (dietary habits scores<4)” was added.

13- (page 15) Every new paragraph in the Discussion should start with your findings rather than the findings of someone else

Thank you, the sentence “Obesity has been linked to poor dietary habits. 8” was transferred to the end of the paragraph.

14- (page 15) What is the need of mentioning prevalence of overweight/obesity in other arab countries here.

Thank you, we wanted to compare our findings in Saudi Arabia, with previous findings in regional countries including the Gulf countries e.g., UAE and other Arab countries e.g., Jordan and Egypt.

15- (page 15) Data given in this para is not gender specific while justification is gender based. Justify?

Thank you, the overall conclusion from reference (35) supports our findings that practicing physical activity is low, and the reference add an explanation and a consequence for this finding. So, however our results were not gender specific because our study main objective was to compare eating habits not physical activity, but reference (35) supported our finding regarding physical activity.

16- (page 15) How does this claim justify the objectives?

Thank you, eating regular meals is a part of healthy eating habits, while our finding revealed that only the minority of students reported eating regular meals. For more clarification, the paragraph was modified into “Our study revealed that only 37.1% and 39.2% of students at health and non-health colleges, respectively reported eating regular meals, which was comparable to …”

17- Generalized justification which lack scientific evidence.

Thank you, the results of references 41 and 42 support this justification.

18- Everywhere in the discussion data is compared with other countries. while scientific justification of the findings are missing.

Thank you, this cross-sectional study aimed to compare and identify eating habits among university students. Comparing our results with previous studies in Saudi Arabia and other countries helps to identify differences in eating habits among students from different contexts. For justification the following sentence was added “Fast food is convenient, easily accessible, and often fits students’ busy schedules. Additionally, peer influence plays a significant role as eating at fast-food chains is a social activity among young people besides the high frequency of skipping breakfast among studied students. Also, fast food is marketed heavily towards teens and young people”

---

## [Editor Report · Decision Letter 1]

14 Oct 2024

Eating Habits of Students of Health Colleges and Non-Health Colleges at ‎the Northern Border University in the Kingdom of Saudi Arabia

PONE-D-24-23613R1

Dear Dr. Bayomy

We’re pleased to inform you that your manuscript has been judged scientifically suitable for publication and will be formally accepted for publication once it meets all outstanding technical requirements.

Kind regards,

Mohammad Sidiq, PhD Pain Sciences Physiotherapy

Academic Editor

PLOS ONE

Additional Editor Comments (optional):

I am satisfied with the revision and i propose it to be accepted for publication
---

## [Editor Report · Acceptance letter]

17 Oct 2024

PONE-D-24-23613R1 

PLOS ONE

Dear Dr. Bayomy, 

I'm pleased to inform you that your manuscript has been deemed suitable for publication in PLOS ONE. Congratulations! Your manuscript is now being handed over to our production team.

Kind regards, 

on behalf of

Dr. Mohammad Sidiq 

Academic Editor

PLOS ONE